# Psycho-Behavioural Changes in Dogs Treated with Corticosteroids: A Clinical Behaviour Perspective

**DOI:** 10.3390/ani12050592

**Published:** 2022-02-26

**Authors:** Lorella Notari, Roxane Kirton, Daniel S. Mills

**Affiliations:** 1Royal Society for the Prevention of Cruelty to Animals (RSPCA), Horsham RH13 9RS, UK; roxane.kirton@rspca.org.uk; 2School of Life Sciences, University of Lincoln, Lincoln LN6 7TS, UK; dmills@lincoln.ac.uk

**Keywords:** affect, aggressiveness, agitation, anxiety, behaviour, corticosteroids, dog, emotion, stress

## Abstract

**Simple Summary:**

Corticosteroids hormones are central to stress responses and, along with other hormones and neurotransmitters, contribute to the onset of physiological and behavioural changes aimed at helping the animal cope with anticipated demand. Both the human and animal literature suggest that exposure to systemic exogenous corticosteroid treatments can be associated with negative emotional states. In this paper, the potential behavioural effects of exogenous corticosteroid treatment on dogs and other species are discussed to show why consideration should be given to this matter when prescribing these drugs.

**Abstract:**

Arousal and distress are often important factors in problematic behaviours, and endogenous corticosteroids are important mediators in the associated stress responses. Exogenous corticosteroid treatments have been reported to change behaviour in human patients and laboratory animals, with similar changes also noted in pet dogs. These behaviours not only potentially adversely impact the welfare of the dogs, but also the quality of life of their owners. Indeed, corticosteroids can bias sensitivity towards aversion in dogs. A decrease in behaviours associated with positive affective states, such as play and exploratory behaviours, together with an increase in aggression and barking have also been described in dogs. According to the available literature, human patients with pre-existing psychiatric disorders are more at risk of developing behavioural side effects due to corticosteroid treatments. It is reasonable to consider that the same may happen in dogs with pre-existing behavioural problems. In this paper, the possible behavioural side effects of exogenous corticosteroids are summarised to help inform and support veterinarians prescribing these drugs.

## 1. Introduction

Arousal and distress are important factors when dogs show problematic behaviours. The regulation of stress reactivity is a process that ensures healthy adaptation to increased anticipated demands, facilitating a return to a state of physiological and psychological balance, or homeostasis. The dysregulation of this process greatly influences the onset and worsening of both physical and mental problems [1].

The hypothalamic-pituitary-adrenal (HPA) axis and the sympathetic-adrenal-medullary (SAM) axis are the major neuroendocrine regulators of homeostasis in vertebrates. They are essential for the control of neural, endocrine, and immune responses to challenges [2]. Corticosteroids and catecholamines are the main chemical messengers in these pathways. The differential responses of these two hormonal systems depend on the type and intensity of the stressor and how it is perceived and interpreted by the animal. Several studies have shown that there is an individual vulnerability to stress [3,4,5].

There are two types of corticosteroids produced by the adrenal cortex, mineralocorticoids and glucocorticoids, and they mediate distinct physiological responses. Aldosterone is the major mineralocorticoid and regulates sodium homeostasis (hydration stress), while the main glucocorticoid varies with species, their primary function is the mediation of a wider range of energetically demanding stress responses.

In this paper, we address the possible behavioural side effects of corticosteroids in dogs, and we use the term “corticosteroid” to refer to glucocorticoids, especially cortisol in the dog. Their effects are mediated by both mineralocorticoid receptors (MRs) and glucocorticoid receptors (GRs) in brain areas that are crucial for memory, learning, and emotion. Under baseline conditions, MRs are largely occupied; however, under stressful conditions, they become saturated and the occupation of GRs increases (Figure 1).

MRs in the hippocampus are particularly important for memory, but this is indirectly modulated by GRs which are abundant in hypothalamic corticotrophin-releasing hormone (CRH) neurons and in the pituitary. When MRs are predominantly activated, hippocampal neurons receive excitatory activation while the additional activation of GRs impairs hippocampal transmission. When either glucocorticoid levels are low and MRs only partially activated or glucocorticoid levels are very high, leading to GRs being predominantly activated, memory and learning may be impaired [6,7]. Glucocorticoid levels affect many other key brain structures that are crucial for emotional and cognitive processes, such as the limbic system and prefrontal cortex. These structures are profoundly interconnected with each other and the hippocampus and can thus be modified by stress causing associated behavioural responses [8]. Physiological stress responses are indispensable for dealing with life’s challenges but should be followed by an individual returning to the previous steady state. Corticosteroids are essential for protecting against adverse events, but their effect can turn from being adaptive to maladaptive in cases of very intense, prolonged, and repeated stimulation in an environment in which an animal has limited control. Maladaptive behaviours, such as an increase in avoidance strategies and aggression, can appear in contexts where adaptation is difficult or impossible from an animal’s point of view [9,10]. In laboratory animals, it has been reported that exposure to exogenous corticosteroids may increase the memory of fearful events, decrease cognitive flexibility, and increase anxiogenic-like behaviours [11,12].

Exogenous corticosteroids are synthetic analogues of natural steroid hormones, and they are largely used in both human and veterinary medicine, mainly for their anti-inflammatory and immunosuppressive effects. The synthetic analogues of cortisol have increased glucocorticoid activity and there are different potencies in different molecules (Table 1) [13]. In the human field, there is growing evidence that higher potency corticosteroids can influence the severity of psychiatric side effects and switching to lower potency corticosteroids may decrease these side effects [14,15].

The wide use of corticosteroids in veterinary medicine and the scarce reporting of behavioural changes compared with other more frequently reported side effects might indicate that either these changes are rare/eligible, or that they were not considered to be related to the treatment and therefore not reported to the clinician [16]. Another possible explanation could be that either owners do not feel they need to report signs of changes in their dogs’ behaviours to their vet or vets do not inquire about their patients’ behaviour. However, given the literature on this topic in humans, a careful review of the veterinary literature is timely. A few studies on veterinary behaviour have specifically investigated the association between corticosteroid therapy and behavioural changes in dogs and the findings of these studies, along with the reports of behavioural and psychiatric side effects in human and laboratory animal studies, indicating that exposure to corticosteroid treatment can have generalisable effects on the behaviour and cognition of mammals [17,18,19,20,21,22,23,24].

Veterinary surgeons have a duty to inform their clients about any possible side effects of the medications that they prescribe and, in the case of corticosteroid treatment, disseminating knowledge of the existence of possible behavioural side effects, including aggression, will protect the welfare of dogs and the safety of both their owners and the wider public. Accordingly, we describe the main possible behavioural changes in dogs treated with corticosteroids below, so that vets have a point of reference to enable them to make an informed professional judgement on the relative costs versus benefits of prescribing these medications in a given case.

## 2. Behavioural Changes in Dogs Exposed to Exogenous Corticosteroids

Assessment of the behavioural changes associated with environmental, social, medical, or pharmacological influences poses several challenges, as the observable animal behaviours are a result of complex physiological and pathophysiological mechanisms. Each individual responds in different ways to different external or internal “triggers”, and the same individual can respond in a different way to the same trigger depending on other concomitant internal or external circumstances. The reported behavioural changes in subjects exposed to corticosteroids are most parsimoniously explained in terms of signs of a negative affective state and reduced welfare that may increase the risk of stress-related behavioural problems, such as aggressive behaviour in certain circumstances [18].

The behavioural changes associated with exogenous corticosteroid treatments in dogs have only been reported in a few studies, but given the available evidence from other species and widespread use of systemic glucocorticoids in veterinary medicine, these possible changes should be recognised and acknowledged by practicing veterinarians.

### 2.1. Increased Vigilance and Agitation

Treatment of human patients with exogenous corticosteroids has been shown to cause a range of psychiatric disorders, including hypervigilance, agitation, and irritability [25,26]. The brain is constantly exposed to corticosteroids and there is a natural fluctuation through a 24 h period as well as peaks that occur as a consequence of exposure to stressful situations. After a surge of corticosteroids, there is an increase in attention and vigilance. Cognitive activities link the stressful event with contextual elements, and the balance between emotional and cognitive processes reflects a strategy to adapt to the changes and associated challenges [27]. In humans and rodents, it has been shown that, for a period of up to 20 min after a peak of cortisol due to a stressor or exogenous sources, emotional processing, distraction, and vigilance are enhanced [28]. Neonatal rats exposed to dexamethasone, a synthetic corticosteroid, have also shown increased startle reactivity [29].

Hypervigilance is a behavioural sign of fear, anxiety and certain other forms of stress. Hypervigilant dogs are more prone to focus their attention on potential threats [10,30,31]. The consequences of this should be considered when dogs are treated with corticosteroids. One survey of owners has reported that dogs being treated with prednisolone become more vigilant and prone to startle, and it has been suggested that the long-term demand of hypervigilance can have negative psychological effects [16].

High arousal and hypervigilance may also be associated with pain-related aggression in dogs, which might lead to corticosteroid prescription [32]. Without careful assessment of timelines, it can be difficult to distinguish the role of pain and possibly increased endogenous corticosteroids from the influence of treatment in the onset of aversive behavioural responses, and the possible cumulative effects of the two cannot be excluded.

Dogs on corticosteroids may show increased fear-related behaviour and avoidance of contact with people. This may include a higher tendency to withdraw when approached or attempts to snap or bite in this circumstance. If the owner describes their dog as being shy, with a tendency to withdraw when approached, it is important that they ask unfamiliar people not to come too close (we suggest about 2 m, as this seems to delimit a personal space boundary) [33] or touch their dog (another critical threshold for triggering an aggressive response) [34] and that they are particularly careful around children. In the same way, if a dog is sensitive to sounds, the owner must limit exposure to unusually noisy environments during corticosteroid therapy. Fearful dogs might worsen symptomatically during this time [18] and owners should be advised to keep them in quiet and protected places, again avoiding contact with children and strangers, if possible. Within the home, a double door separation policy should operate to keep dogs away from young children, as this minimises the risks from a door being accidentally left open in a busy domestic environment. For dogs that show severe signs of fear, professional behavioural support should be sought. A summary of behavioural signs, examples of possible situations, possible outcomes, and recommended actions are presented in Table 2.

### 2.2. Increased Barking

Barking is part of the normal behavioural repertoire of dogs. Dogs bark in different contexts and with different motivations, intensities, and frequencies. Increased barking not related to play or attention-seeking contexts is often related to stressful situations, and barking levels, along with other vocalisations, have been used as part of welfare assessments in dogs [35,36,37].

Excessive barking is also an undesirable behaviour that is often complained about by dog owners and their neighbours, and it is a frequent reported behavioural sign of separation-related problems [38,39,40]. It has been reported that dogs receiving corticosteroid treatment barked significantly more than when they were not receiving therapy [17], and an increased tendency to bark can be observed in dogs that are fearful or distressed [41] Considering that barking, when intense and protracted, can be a sign of poor welfare and is perceived by the owner as a problem, owners should be warned of this potential side effect when their dogs are on corticosteroid treatment.

Excessive barking as a side effect of corticosteroid therapies can be effectively managed and should not be simply suppressed through the application of aversive stimuli [42]. It is important that the veterinarian ask the owner to describe the contexts where the dog tends to bark. If barking is mainly motivated by alarm or territorial defence, the owner should be advised to keep their dog away from the property boundaries and, in general, to try to shelter them from any triggering stimuli, with a view to preventing a worsening of this behaviour. If barking is mainly motivated by attention seeking, the owner should be advised on how to manage this behaviour without providing inadvertent reinforcement through differential reinforcement of alternative behaviours [43]. It should be noted that reinforcement can include attempts to tell the dog off. Positive reinforcement when the dog is calm should be increased to produce a favourable shift in the time budget. As bringing about change in owner behavioural habits can be challenging, professional advice is indicated if the owner is unable to manage the situation effectively and humanely. A summary of behavioural signs, examples of possible situations, possible outcomes, and recommended actions are presented in Table 3.

### 2.3. Aggressive Behaviour

In both humans and non-human animals, it has been shown that stress reduces socio-positive behaviours and increases aggression [44,45,46]. The possible effects of exogenous glucocorticoid treatment are difficult to predict, can vary according to the patient’s profile and are often dose dependent. In human medicine, it has been reported that higher doses and long-term therapies are more at risk of contributing to the development of neuropsychiatric symptoms, including anxiety, depression, and aggression [44,45] Aggressive behaviour towards conspecifics has also been shown after corticosterone infusion in the brain ventriculi of rats with a response being stimulated rapidly after infusion [46].

It has been reported by owners that dogs receiving treatment with corticosteroids showed a tendency to react aggressively when petted or even just approached and that their dogs also appeared to be significantly more prone to avoiding people or situations [17]. It was also found that dogs exposed to corticosteroids after being referred for behavioural problems were significantly more likely to show behaviours motivated by negative affective states compared with dogs not exposed to these drugs [18].

The affective state of animals, including humans, creates certain behavioural tendencies, and it has been suggested that prednisolone, a synthetic corticosteroid, may intensify the brain’s response to negative emotional situations and increase irritability through amygdala over-reactivity [47,48]. The risk of increased aggression in such circumstances is greater in individuals who already have a predisposition towards aggressive behaviour. It has been suggested that this risk might be mediated by a corticosteroid related change in the regulation of the serotonergic system in subjects with pre-existing psychiatric disorders [49,50,51,52,53] The role of serotonin in the modulation of impulsivity and aggressive behaviours has been reported in both humans and other animals, including dogs, and it has been hypothesised that serotonin is heavily involved in pro-social behaviours [54,55].

In line with the human literature [56], it seems reasonable to suggest that the risk of behavioural side effects from these drugs is likely to be more severe in dogs with pre-existing behavioural problems, and owners should be informed and fully advised of the associated risks accordingly. The fact that corticosteroids can increase appetite should also be considered, and dogs in treatment with these drugs may be more motivated to defend food as a result. Veterinarians should advise owners to implement a regular and safe feeding routine in order to avoid situations that may create unexpected reactions, such as when someone comes close to the dog’s bowl or when the dog is approached while they are eating. Food should be given in a quiet area of the house, and the empty bowls removed when the dog has finished. Children should be supervised to ensure that they do not go near the dog when there is food or chew-toys nearby [57,58] In general, it is best if owners with dogs with previous aggressive behavioural problems in the presence of food are advised to seek professional behavioural advice to address this problem and the associated risks more fully in both the short and longer term.

As dogs receiving treatment with corticosteroids may become more irritable, and this tendency may also be related to some form of physical discomfort for which the corticosteroids have been prescribed, it is important to avoid unpredictable physical contact. The owner should be advised to ask their dog to come towards them for interaction, rather than going to the dog. The veterinarian should recommend instructing all family members to do the same, especially if there are children in the home. It is also important to clearly recommend that children avoid hugging or holding their dog while they are not well and/or receiving therapy with corticosteroid drugs. If the dog growled when touched in the past, access to children and strangers should be prevented as a precaution, using a double door system. If the dog has growled in the past when being groomed, grooming should either be avoided during treatment or, if this is not practical, the dog should be muzzled prior to grooming. Pre-existing irritable aggression problems should be addressed appropriately by advising the owner to seek professional behavioural advice. A summary of behavioural signs, examples of possible situations, possible outcomes and recommended actions are presented in Table 4.

### 2.4. Decreased Play and Exploratory Behaviours

Play is considered a positive welfare indicator in many species, including humans. When an animal seeks opportunities to engage with potentially rewarding experiences, it can be assumed that their welfare is improved by such activities. In the same way, exploratory behaviour can also be a positive sign of seeking rewarding experiences, while decreases in these behaviours have been associated with negative affective states, as the animal’s desire to seek new information is reduced, and avoidance is more likely [59,60]. Dogs on exogenous corticosteroid treatment were significantly less exploratory in a behaviour test [18] and a survey conducted with dog owners reported that dogs on corticosteroids were significantly less playful when in treatment with these drugs [18]. Reduced play and exploration are not likely to be perceived as problematic behaviours because they may not immediately negatively affect owners unlike the occurrence of aggressive behaviours or excessive vocalization [61,62]. Owners may not consider these behavioural signs unless informed about them or prompted with appropriate questions.

The circumstances, animal’s history, and personality need to be taken into account when considering the level of play and/or exploratory behaviours as potential indicators of negative affective state. A comparison should be made with previous levels of these behaviours in similar circumstances, in order to gain insight into the animal’s affective state [60,63,64]. When there are behavioural signs that can be linked to negative emotional states, this suggests the welfare of the animal is compromised, and there is an increased risk of other behavioural problems.

### 2.5. Monitoring the Behaviour of Dogs in Treatment with Corticosteroids

Monitoring patients receiving treatment with medications can enhance owners’ compliance and help to detect possible adverse events during therapy. This is important for both welfare and safety reasons. Setting up a monitoring schedule when corticosteroid medications are prescribed will also give the veterinarian the opportunity to discuss the behaviour of the dog with their client, identify any pre-existing problems, and support them with advice.

An example of a monitoring form for dogs in treatment with corticosteroids is provided in Figure 2. In human medicine, electronic forms seem more effective in terms of compliance compared with paper forms [65,66]. Electronic forms can be easily prepared and sent to the client as weekly web links, before and during treatment.

## 3. Discussion and Conclusions

Corticosteroids are routinely prescribed in veterinary medicine but very few studies have extensively analysed the clinical behavioural side effects of these drugs in dogs. One study analysing data from a population of 455,557 dogs under veterinary care reported that 6.2% of the investigated sample received systemic exogenous glucocorticoids. A random sample of 3000 dogs receiving corticosteroids were reviewed in detail and reported side effects were found in 148 (5%) of these dogs. Behavioural changes were only reported in 4% of this subsample (six dogs), and these were all related to aggressive behaviour [67] These numbers would seem to indicate that the behavioural side effects of corticosteroid medications in dogs are rare, but potentially severe. However, as discussed above, this figure may also be a gross underestimate, due to the lack of routine consideration of potential behavioural side-effects. Specific monitoring of behavioural side effects is to be advised, using a simple system such as that illustrated in Figure 1.

The reported case studies in human medicine and studies conducted on laboratory animals provide compelling evidence to assume that similar side effects occur in dogs. In human psychiatry, it has been reported that corticosteroid treatments affect mood, memory, and cognition, and the incidence of adverse effects has been associated with corticosteroid dose, repeated treatments, and pre-existent psychiatric disturbances [68,69]. The impact of the psychiatric side effects of corticosteroids on human patients in the UK has been quantified by analysing longitudinal medical records over an 18-year period. The authors found patients exposed to exogenous corticosteroids were five to seven times more at risk of suicide, delirium, confusion, disorientation, mania, and panic disorders. They also found that high doses of corticosteroids and a prior history of neuropsychiatric disorders were associated with a higher risk of negative psychiatric outcomes [56].

The scarce reporting of the clinical behavioural side effects of corticosteroids in dogs is possibly related to a lack of awareness and the potential difficulty in identifying these side effects and behavioural changes, given that they could often be associated with the disease for which these drugs may have been prescribed. For example, corticosteroid medications are largely used in dermatology, and pruritus can increase irritability, however it was reported that the treatment, and not the pruritus, was associated with an increased reactivity to potentially fearful stimuli [70].

Corticosteroids are undoubtedly useful veterinary medicines, but current evidence of the possible behavioural side effects from using exogenous corticosteroids, along with the importance of giving proper advice to dog owners when these medications are prescribed, needs greater recognition and should be used to inform professional judgements on a case-by-case basis. Indeed, we support the call [71] for a greater awareness within the veterinary community of what medical treatments affect animal behaviour and the importance of monitoring and reporting of behavioural side effects as adverse events.

## Figures and Tables

**Figure 1 animals-12-00592-f001:**
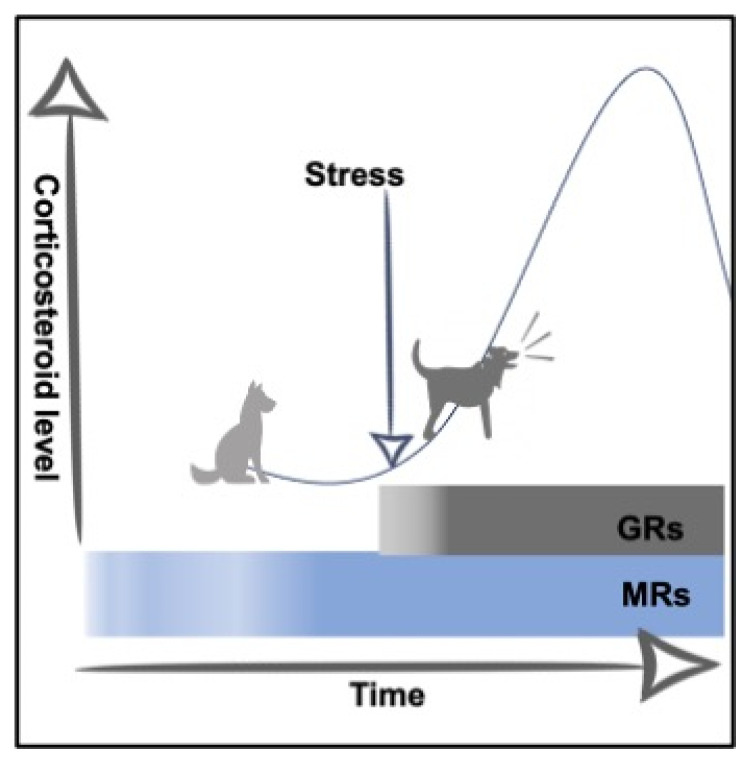
Schematic diagram showing mineralocorticoid and glucocorticoid receptors occupancy in baseline and stressful conditions. GRs = Glucocorticoid Receptors; MRs = Mineralocorticoid Receptors.

**Figure 2 animals-12-00592-f002:**
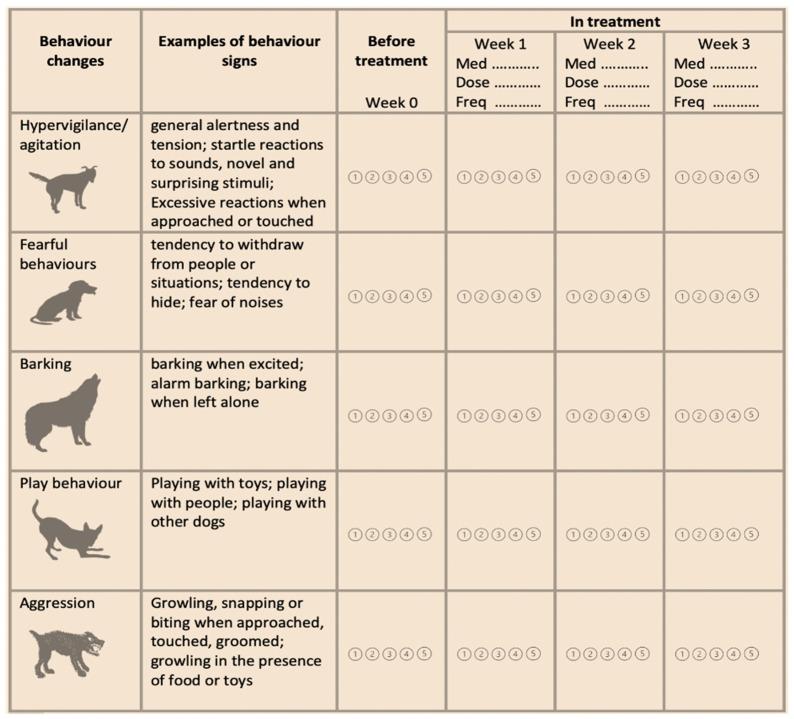
Example of weekly monitoring of dog behaviour when in treatment with exogenous corticosteroids. Owner should be asked to rate the presence of behavioural signs from one to five. 
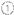
 = Never; 
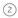
 = Rarely; 
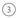
 = Sometimes; 
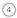
 = Often; 
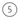
 = Very often; Med. = name of the medication; Dose = dose of the medication; Freq = frequency of administration.

**Table 1 animals-12-00592-t001:** Corticosteroid potency comparisons.

Synthetic Corticosteroids	Equivalent Glucocorticoid Dose (mg)	Anti-Inflammatory Potency Relative to Hydorcortisone	Mineralo-Corticoid Potency Relative to Hydorcortisone	Half-Life Duration of Action (h)
Hydrocortisone	20	1	1	8–12
Cortisone acetate	25	0.8	0.8	8–12
Prednisone	5	4	0.8	12–36
Prednisolone	5	4	0.8	12–36
Methylprednisolone	4	5	0.5	12–36
Dexamethasone	0.75	30	0	36–54

**Table 2 animals-12-00592-t002:** Increased vigilance and agitation during corticosteroid treatment. Behavioural signs, examples of possible situations and contexts, possible outcomes, and recommended actions.

Behavioural Signs	Example Contexts	Risk/Possible Negative Outcomes	Recommended Actions
General alertness and tension. Startle reactions to sounds, novel and surprising stimuli. Excessive reactions when approached or touched. Avoidance of contacts with people. Withdraw from people or situations. Tendency to hide.	When approached or touched. In noisy and busy domestic environments. In unpredictable social situations (e.g., in the presence of strangers, children, other dogs). When there are loud noises (fireworks, thunderstorms).	Reduction of social interactions and worsening of the owner-dog relationship. Hiding and refusing to be handled. Snapping or biting.	Ask unfamiliar people not to touch the dog and not to approach him/her. Keep the dog separated from young children using secure barriers. Limit exposure to noisy environments. In case of severe signs of fear, seek professional behavioural support.

**Table 3 animals-12-00592-t003:** Increased barking during corticosteroid treatment as a behavioural sign with examples of possible situations and contexts, possible outcomes and recommended actions.

Behavioural Sign	Example Contexts	Risk/Possible Negative Outcomes	Recommended Actions
Increased intensity and frequency of barking.	Barking behind doors, fences, gates in response to stimuli (visual or auditory) that predict arrival of people or the approach of another dog. Barking at people or other dogs during walks Barking to get the owner’s attention. Baking when left alone.	Use of inhumane methods to stop the barking with negative consequences for the dog’s welfare. Neighbour complains and local authority interventions. Worsening of separation related behaviours.	Reduce the dog’s exposure to triggering stimuli. Avoid inadvertent reinforcement and consistently reinforce the dog for calm behaviours. Reduce the time the dog is left alone. Seek professional behavioural support if the barking when left alone was pre-existing.

**Table 4 animals-12-00592-t004:** Aggressive behaviour during corticosteroid treatment. Behavioural signs, examples of possible situations and contexts, possible outcomes, and recommended actions.

Examples of Behaviour Signs	Examples of Possible Contexts	Risk/Possible Negative Outcomes	Recommended Actions
Growling Snapping Biting	When the dog is approached in his/her bed. When the dog is petted or hugged. When the dog is approached or touched by children. When the dog is approached by unfamiliar people. When the dog is groomed. When the dog is eating or chewing.	Reduction of social interactions and worsening of the owner–dog relationship. Use of inhumane methods to control aggressive behaviours with negative consequences for the dog’s welfare. Physical harm to people or other animals. Legal issues. Euthanasia.	Allow the dog approach in its own time rather than actively approach him/her. Family children must not hug or pet the dog. Safety measures to separate the dog from children when they cannot be actively supervised. Unfamiliar adults and children must be prevented from approaching and touching the dog. The dog must be muzzled when groomed. The dog must not be approached when he/she is resting, eating, or chewing. Seek professional behavioural support.

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
