# Peer review of "Psycho-Behavioural Changes in Dogs Treated with Corticosteroids: A Clinical Behaviour Perspective"

_animals, 2022, doi:10.3390/ani12050592_

Round 1
Reviewer 1 Report
The present work examines the potential behavioural effects of exogenous corticosteroid treatment on dogs and other species. The present approach derives from the need to assess this impact because these behaviours potentially adversely affect the welfare of the dogs, and also the quality and security of their owners life. The topic is very interesting and attractive: it summarizes the current information about changes of dogs behaviours during corticosteroid treatment, giving some suggestions for veterinarians and owners to manage this situation. I agree on how the authors arranged the manuscript according to themes; the paper is well structured, but I would move the section “2.2. Decreased Play and Exploratory Behaviours”at the end of this chapter (before of the monitoring), to assure a better logical line (vigilance, barking and aggressiveness are correlated with an anxiety status). In general, the references are specific and relevant, but the reference 31 is missing, the references 35 and 36 are the same. Please, correct the bibliography.
Title: It is adequate for the content of the paper.
Summary and Abstract: They are both well written and recap the information contained in the main text without being repetitive.
Keywords: They are proper but you could add more keywords as : “Vigilance”, “Anxiety” , “Agitation” and “Aggressiveness”, being the assumptive side effects of corticosteroid treatment.
Introduction: It is clear and the context and information on the topic explains the aim of the comment.
Chapter 2, Discussion and Conclusions: They follow a logical line and present persuasive interpretations of the available current information about the topic. Also, they are accompanied by a clear figure of an useful monitoring form. In addition, the authors give suggestions to help inform and support veterinarians prescribing these drugs.
The paper is well written and, since it interprets information about Ethology and medical therapy of dogs, it deserves to be published after minor revisions.
Author Response
Dear Reviewer N.1 ,
We would like to thank you for your appreciation of our paper, we addressed all your comments in the attached file
Kind Regards
The Authors

Reviewer 2 Report
In the simple summary there needs to be a little more consideration for the lay reader. For example most people will not know what corts are. So perhaps "Stress hormones (corticosteroids) are important... etc" also "emotional" rather than "affective" maybe? I would suggest reading it again with the layperson in mind.
Unfortunately there are no line numbers...
ref needed at end of intro para 1.
Para 2: I would delete "Stress.... circuits, with" and simply say "the HPA axis and SAM are the major....etc". If you would rather not can you think of a less vernacular term than "brain circuits"?
P2 first line remove "hormones" it's unnecessary (tautological)
P2: is it possible to provide a figure that shows the of GR and MR
P2: whether an individual should return to a "stable state" is situationally dependant. In companion animals this is "ideal". I also assume that the stable state is one of neutral or positive affect? Perhaps make the presumed baseline explicit.
P3 first lines: Is it possible that they are not generally considered at all (rather than not considered to be related)?
P3 section 2: the word "different is used a lot of times, such that the meaning becomes confusing.
I would ask for an extended table with columns that show the situation, the potential outcomes and the recommended responses for dogs/their owners. Although they are in the text a table would help to both elucidate the ideas and provide a helpful reference resource. I would, do this for each section.
End of section 2.2. What other behavioural problems show and increased likelihood? If the treatment is short-term then the probability of long term behavioural effects may be diminished. To what extent were play and exploration reduced? Providing data on the papers you have reviewed allows the reader to see the magnitude (or perceived magnitude) of these changes. At a basic level the larger the change the greater the welfare issues.
2.3: how can the professional/owner distinguish between the different stimuli that promote increased barking (and again, what was the underlying difference that caused the sig diff?)
What are "behaviours motivated by negative affective states"?
Last para pg 5 requires referencing
As above, I think the major omission her is a clear guide for veterinarians as part of the commentary. Most professionals may not have the time to read the manuscript and make their own guidance. This would reduce the impact of the work
Author Response
Dear Reviewer N.2,
Thank you for your comments and suggestions, we have addressed all of them in the attached file
Kind Regards
The Authors

Round 2
Reviewer 2 Report
Thanks to the authors for their responses. I am happy that they have addressed the issues raised by the reviewers.
Just 1 or 2 very minor points:
L1: remove the last "s" from "corticosteroids"
P19 L18: "effectively managed through neutral or positive interventions, without the need for aversive stimuli or punishment"